# Connections between Family Assets and Positive Youth Development: The Association between Parental Monitoring and Affection with Leisure-Time Activities and Substance Use

**DOI:** 10.3390/ijerph17218170

**Published:** 2020-11-05

**Authors:** Maider Belintxon, Alfonso Osorio, Jokin de Irala, Marcia Van Riper, Charo Reparaz, Marta Vidaurreta

**Affiliations:** 1Department of Community, Maternity and Pediatric Nursing, School of Nursing, Campus Universitario, Universidad de Navarra, 31009 Pamplona, Spain; mbelintxon@unav.es (M.B.); mvidaurreta@unav.es (M.V.); 2IdisNA, Navarra Institute for Health Research, Irunlarrea 3, 31008 Pamplona, Navarra, Spain; jdeirala@unav.es; 3Institute for Culture and Society, Campus Universitario, Universidad de Navarra, 31009 Pamplona, Spain; 4School of Education and Psychology, Campus Universitario, Universidad de Navarra, 31009 Pamplona, Spain; creparaz@unav.es; 5Preventive Medicine and Public Health, School of Medicine, Campus Universitario, Universidad de Navarra, 31009 Pamplona, Spain; 6School of Nursing and Carolina Center for Genome Sciences, University of North Carolina, Chapel Hill, NC 27599, USA; vanriper@email.unc.edu

**Keywords:** adolescents, health assets, positive youth development, constructive leisure activities, substance use, lifestyles

## Abstract

This study aimed to determine the associations between parental monitoring and affection and three adolescent lifestyle aspects: constructive leisure, non-constructive leisure and substance use. A cross-sectional study was conducted in four countries (Chile, Mexico, Spain and Peru). Adolescents aged 12–15 self-completed a multi-purpose questionnaire. Multiple logistic regressions were performed to analyse the association between the parental monitoring and affection variables and the outcomes in terms of the children’s lifestyles. The results indicate that parental monitoring is conducive to more constructive leisure and less non-constructive leisure and seems to be conducive to the prevention of substance use. Furthermore, parental affection is conducive to constructive leisure and the prevention of substance use. The discussion focuses on the fact that the family can be a protective resource associated with positive adolescent development.

## 1. Introduction

Historically, the adolescent stage has been considered a turbulent and conflictive growth period [1,2,3] characterised by problems and submersion in an inevitable biological storm [4]. In the study of adolescence, a deficit-focussed model has predominated [2,4,5].

Deficit models focus on pathology and identify the problems and imbalances of this growth stage [5]. However, since the end of the 20th century, a change of focus has been observed in development theories [4]. This change in approach has led to a new model focussed on positive development and competence during adolescence [5,6,7]. This new approach holds that an adequate transition to adulthood requires more than the avoidance of risky behaviours and requires the achievement of evolutionary achievements [2]. However, the positive development model and the deficit model are complementary models, since reducing and preventing deficits and behavioural problems and promoting development and competence are parallel paths.

The “positive youth development” (PYD) model considers the adolescent as an individual in a period of psychological, emotional, social and intellectual growth [3,4]. This approach is rooted in the theory of relationship development, which indicates that human development is not predetermined, is probabilistic and is relatively plastic, since there is always the possibility of change [7]. This plasticity leads to a complex system of relationships between individuals and their contexts [2,8,9]. The adolescent needs to be involved in relationships and contexts that facilitate and promote their optimal development and a correct transition to adulthood [3,9,10]. The relationships between the adolescent and their context constitute the basis of behavioural and personal development.

The PYD model is based around the concept of resources or assets for development [2]. Specifically, the Search Institute identified 40 assets that promote positive development [5,9,11,12,13]. Health assets refer to personal, family, school or community resources that provide the support and experiences necessary for positive development during adolescence [14]. These 40 assets are 20 internal assets and 20 external assets [13] that mobilise the person towards healthy behaviours and welfare outcomes [15,16]. External assets refer to the characteristics of the family, school or community [2] in which the adolescent lives, and internal resources refer to psychological or behavioural characteristics of the adolescent, such as high self-esteem, personal responsibility, future expectations and decision-making ability [9,10].

In this sense, the literature indicates that the functional family is a protective factor. It can directly affect adolescents and increase the probability they will obtain positive health results [2,9,17,18]. In fact, together with the school and the community, it is one of the most important external assets to the adolescent and has a strong influence on the acquisition of positive health behaviours [12,19].

Family assets include important factors connected to family, such as family support, positive communication, clear rules and consequences and supervision [9]. These variables may also influence the adolescent’s choice of activities, greater psychosocial well-being and lower risk of adverse health behaviours [20,21].

The family and family functioning play an important role in the types of leisure activities performed by adolescents [17]. Leisure time has an impact on adolescent development because it represents a place of free time and meaningful choices. Social scientists distinguish two types of leisure activities. On the one hand, constructive leisure is composed of structured activities that require a long-term commitment (such as belonging to sports or religious clubs). It has a protective effect against the avoidance of risky behaviours. On the other hand, non-constructive leisure includes unstructured activities such as watching TV or surfing the Internet, and it is more related to risky behaviours [18,19,22,23]. The family is also a protective factor against substance use [9]. Substance use is one of the highest current prevalent adolescent risk behaviours and a major global public health concern [24,25]. Parents have an opportunity to play an important role in preventing their adolescent children from engaging in risky behaviour and promote positive behaviours [26]. This transitional period is of considerable practical interest since many youths begin getting involved in substance use [27] and risky behaviours. Furthermore, the literature shows that substance use during early adolescence is associated with a diverse range of problematic physical, mental health and social issues in later adolescence and adulthood [27,28].

Previous studies have analysed the associations between these adolescent behaviours and different variables, including parental monitoring and affection [17,27]. However, few of these have done so in Spanish-speaking countries. Furthermore, most studies have focused on a deficit-prevention model, disregarding positive adolescent development.

Therefore, this study had the following objectives:▪To examine how monitoring and parental affection differ according to sex, age and other sociodemographic variables of adolescents.▪To confirm the association of monitoring and parental affection with constructive and non-constructive leisure activities using the same sampling and survey tools in an international cohort of four Spanish-speaking countries.▪To confirm the association of monitoring and parental affection with the consumption of substances such as tobacco, alcohol and marijuana in the mentioned international cohort.

## 2. Materials and Methods

To achieve the proposed objectives, the data from an international research project called YOURLIFE were analysed [23,29,30,31,32]. The main objective of this project is to identify the opinions, knowledge and attitudes of high school students about their lifestyles and various aspects related to sexuality and intimate relationships, as well as the factors that influence them. This project is ongoing, uses self-administered on-line questionnaires and has both cross sectional and longitudinal data. The dataset is made up of more than 24,000 students from 10 countries.

The article presented here is framed under a cross-sectional design covering four countries (Chile, Mexico, Spain and Peru).

### 2.1. Sample

A convenience sampling was carried out in public and private schools in Chile, Mexico, Spain and Peru. Invitations were sent out, and 52 schools voluntarily decided to participate. An online questionnaire was completed by 3443 secondary school students of approximately 13 years of age. After eliminating those who did not indicate their age or sex and those who were outside the age range of 12–15 years, a sample of 3300 students remained.

### 2.2. Questionnaire and Variables

For data collection, one of the three questionnaires of the YOURLIFE project was used: The c13 questionnaire aimed at students between 12 and 15 years (https://proyectoyourlife.com/index.html). The questionnaire is anonymous, self-completed and answered online. It is a multi-purpose questionnaire with questions of various types. The items used in the present study are described below.

#### 2.2.1. Independent Variables: Parental Monitoring and Affection

Parental monitoring and affection were measured through the Escala de Educación Familiar (EEF, Family Education Scale) [33]. This instrument assesses parenting styles (monitoring and affection) and education in values (fortitude and privacy). We used the first two subscales, which are described below. The mentioned study showed results which support the instrument’s validity and reliability. In our data, Confirmatory Factor Analyses showed good fit indices (RMSEA = 0.045, CFI = 0.979, TLI = 0.974), confirming the validity of the proposed structure. Reliability data are given below.

##### Parental Affection

The questions regarding parental affection were formulated as follows: Regarding your parents: “Do they know you well and understand you?”; “Do they set an example?”; “Do they listen to you?”; “Do they take your opinions into account when doing something?”; “Do they speak kindly to you?”; “Do they help you when you feel insecure?”; “Do you feel that they love you and that they accept you as you are?”; “Do you feel comforted and supported by them?”; “Do you feel that your things interest them?”; “Do they take time to talk to you?”; and “Do they try to be with you and help you?” Each question had five possible answers (from 0 = not at all to 4 = very much). The mean of the scores of all the items was calculated, and we eliminated the participants who had responded to fewer than half of the items. This variable (parental affection) was dichotomised around the median into “high affection” and “low affection”. Internal consistency was high (Cronbach’s alpha = 0.95).

##### Parental Monitoring

The questions regarding parental monitoring were formulated as follows: Regarding your parents: “Do they require you to follow a schedule?”; “Do they decide with you what you must do?”; “Do they limit what you spend?”; “Do they limit the time you can watch television?”; “Do they control your use of cell phones or the Internet?”; and “Do they control your books and magazines?” Each question had five possible answers (from 0 = not at all to 4 = very much). Again, the mean of all the scores was calculated, and the participants who responded to fewer than half of the items were eliminated. This variable was dichotomised around the median into “high monitoring” and “low monitoring”. Internal consistency was acceptable (Cronbach’s alpha = 0.74).

#### 2.2.2. Dependent Variables: Constructive Leisure, Non-Constructive Leisure and Consumption of Toxic Substances

##### Constructive and Non-Constructive Leisure

The questionnaire included questions to determine the frequency with which students participate in different leisure activities. For each activity, a response scale from 0 to 4 was established (0 = never, 4 = 3 or more days a week). The leisure activities were divided into two groups: “constructive leisure” and “non-constructive leisure”. The election of activities and the division into constructive or non-constructive leisure was based on previous literature [34]. We have already used these variables in other studies [23,35].

Constructive leisure included the following questions: “In the last 12 months, how often have you done the following activities?”: “Play some sport, go to the mountains, etc.”; “Volunteer (collaborate with a non-governmental organisation, charity, etc.)”; “Attend artistic and educational activities (music, painting, theatre, courses, talks, catechesis, etc.)”; and “Activities with your parents (play sports, outings or excursions, play board games)”.

Non constructive leisure included the following questions: “In the last 12 months, how often did you perform the following activities?”: “Hang out in the street, in a park, on the beach, or in other public places”; “Go to shopping centres, game rooms, billiards, or football stadiums”; and “Gather in a place with friends, without adults present”.

For each group of activities, the mean of all the corresponding item scores was calculated and the participants who responded to fewer than half of the items were eliminated. Again, each variable was dichotomised into: “high and low frequency of constructive leisure activities” and “high and low frequency of non-constructive leisure activities”.

##### Consumption of Toxic Substances

Participants were asked how often they consumed various substances, such as alcohol, tobacco, marijuana and other drugs (Never, Fewer than 1 day a month, 1-3 days a month, 1-2 days a week and 3 or more days a week). We decided to create a new variable separating those who answered “never” for all of these substances from those who had ever consumed any of them. This decision was based on the low prevalence of consumption at the ages covered by our analysis and on scientific evidence that indicates that any consumption of these substances during adolescence is considered a risk behaviour [36].

#### 2.2.3. Covariates

The multivariate analyses adjusted for age, sex, educational level of the parents, family structure and religiosity.

Religiosity was assessed with three different variables. Participants were asked what religion they belonged to. Then, if they belonged to any religion, they were asked how often they went to the church/temple of their religion and how often they prayed (from 0 = Never to 5 = More than once a week). They were finally asked how much they agreed with this statement: “My faith is an important influence in my life, and I am willing to take it into account in my decisions” (from 0 = Strongly disagree to 4 = Strongly agree). For each of these issues, a dichotomous variable was generated into “high” and “null/low” attendance/prayer/salience.

### 2.3. Procedure

Complete details of project YOURLIFE and its procedures are available elsewhere [29]. Briefly, in each country, there was a collaborator who was in charge of establishing a personal contact with schools and giving them the information and documentation they needed and/or requested. Schools had thus the information they needed to inform the parents about the study and to carry out the study. Each school handled the request for parental consent based on their own local policies of action [37]. The answers to the study questionnaire were only accessible to researchers and never to parents or teachers. Throughout the process, the student privacy was protected, and there was no possibility of identifying any student who replied to the questionnaire. The students voluntarily answered to the survey. They were informed that they could leave the survey at any moment or leave any question without a response. The research was approved by the Research Ethics Committee of the University of Navarra.

### 2.4. Data Analyses

The characteristics of the participants are provided as absolute frequencies and percentages by country. To determine which variables were associated with affection and parental monitoring, we first analysed the bivariate associations between each of these two variables and various sociodemographic variables (age, sex, family structure, parental education and religiosity). Second, two logistic regressions were performed. The dependent variable was parental monitoring in one case and parental affection in the other. In both regressions, the independent variables were the aforementioned sociodemographic variables.

Then, the associations between the family variables (monitoring and parental affection) and the three outcomes in terms of the children’s lifestyles (constructive leisure, nonconstructive leisure and substance use) were analysed. The two family variables were divided into quintiles. For each quintile, the percentage of participants performing each of the three behaviour outcomes was calculated to perform a preliminary description of these associations.

Three logistic regressions were then performed, each with one of the three lifestyle outcomes as a dependent variable. The independent variables were affection and monitoring, in addition to various sociodemographic variables to adjust for possible confounding.

Finally, we performed a structural equation model (SEM) to test all associations in a single model.

The data were analysed in the statistical programme Stata 12.1. The significance level was set at 0.05.

## 3. Results

In total, 3443 students from Spain, Mexico, Chile and Peru were recruited. In total, 89 participants were excluded because they presented missing data in the age variable or because they reported being younger than 12 years or older than 15 years. In total, 54 participants who did not indicate their sex were also eliminated. Finally, a sample of 3300 students was analysed (393 from Chile, 964 from Spain, 1089 from Mexico and 854 from Peru).

The main characteristics of the sample are described in Table 1. The sample was mostly composed of women (54.9%), and over half of the participants (54.7%) were 13 year olds. Most of the participants lived with their father and mother (83.1%), and at least one of the parents had completed a university degree (76.7%). Most of the students referred to themselves as Catholics (79.5%) but had little or no religiosity when classified using our composite variable of religiosity (66.3%). Half of the participants were in the lower category of monitoring (53.7%) and affection (51.3%). The majority of participants infrequently performed both constructive leisure activities (57.2%) and non-constructive leisure activities (56.1%). 84% of the participants had not consumed toxic substances.

To achieve the first objective, the bivariate and multivariable associations of monitoring or parental affection with various sociodemographic variables were explored (Table 2). After adjusting for possible confounding, boys, younger adolescents, Peruvians and Mexicans compared to Spaniards and those with high values in the religiosity variables reported more frequently having received a higher degree of monitoring by their parents. Parental affection was more frequently reported as being high, after multivariate adjustments, among Spaniards compared to Peruvians and Chileans, among those who lived with both parents, among those who had parents with a university education and among those with high values in the religiosity variables.

To work towards the two other objectives, a bivariate analysis was first carried out to determine the association between each parental variable (monitoring and affection) and each outcome (constructive leisure, non-constructive leisure and substance use) (Figure 1 and Figure 2). Participants who reported higher levels of parental monitoring were involved more frequently in constructive leisure activities and less frequently in substance use, while the association with non-constructive was less clear (Figure 1). A similar pattern was found regarding the association between parental affection and the outcomes (Figure 2).

After these first analyses, multiple logistic regressions were carried out to estimate the associations between the monitoring and parental affection variables and the different outcomes (Table 3). First, the associations with constructive leisure were explored. Both monitoring (OR: 1.49; 95% CI: 1.25–1.79) and affection (OR: 1.52; 95% CI: 1.26–1.82) were independently associated with a higher frequency of constructive leisure activities. Other variables associated with a greater frequency of constructive leisure activities were being Mexican (compared to being Spanish), being Spanish (compared to being Chilean), having parents with university degrees, having a high frequency of church attendance or praying and having high religious salience. 

Second, the association between monitoring and parental affection and non-constructive leisure was explored. Parental monitoring (OR: 0.71; 95% CI: 0.60–0.86) was significantly associated with a lower frequency of non-constructive leisure activities. In contrast, no significant association was found between affection (OR: 0.94; 95% CI: 0.79–1.13) and the frequency of non-constructive leisure. Other variables associated with a lower frequency of non-constructive leisure activities were being younger, being Peruvian (compared to being Spanish), having parents without a university education and having a high frequency of church attendance.

Finally, the association between parental monitoring and parental affection and substance use was studied. Although the association between monitoring and substance use was significant in the univariate analyses (*p* < 0.001) (Table 3), it was on the limit of significance in the multiple regression (OR: 0.84; 95% CI: 0.65–1.08). Affection was associated with a lower prevalence of substance use (OR: 0.49; 95% CI: 0.38–0.64). Other variables associated with lower consumption were being younger, being female, being Peruvian (compared to being Spanish), living with both parents and having a high frequency of church attendance.

The structural equation model obtained excellent fit indices (RMSEA = 0.036, CFI = 0.973, TLI = 0.965). The results are shown in Figure 3. Both monitoring and affection predict more constructive leisure and less substance use. Regarding non-constructive leisure, monitoring seems to prevent it, while affection seems to promote it.

## 4. Discussion

### 4.1. Constructive Leisure

The data obtained suggest that both monitoring and parental affection are associated with a greater frequency of constructive leisure activities. Constructive leisure activities are associated with more positive adolescent development and promote better health outcomes [1,17,20,38,39]. The present study is the first to associate parental control and affection with various constructive leisure activities, in an international sample with diverse cultural values. 

The results of this article are consistent with the theory of parental socialisation. This theory indicates that, when parents invest more hours in structured activities with their children, they dedicate more time to this type of activity [40]. Doing family activities and having greater parental monitoring are associated with constructive leisure activities [17]. Parents who establish open communication and base their relationship on affection promote free-time, extracurricular and volunteer activities [17].

The present study, in agreement with the literature, indicates that a higher educational level in parents [41,42] and higher levels of religiosity [35,43] promote greater participation in activities of constructive leisure. Unlike others [44], this study did not find an association between family structure and a greater participation in structured leisure activities. There was also no association between sex or age and the participation in constructive leisure activities, as found by others [17].

### 4.2. Non-Constructive Leisure

The results show an association between parental monitoring and less non-constructive leisure activities, but no clear association was found between parental affection and this type of leisure. The engagement in non-constructive leisure entails a low development of personal skills and lower levels of motivation. This promotes a sense of worse well-being in adolescents [45]. This feeling of discomfort and worse well-being favours the adoption of risky behaviours (delinquency or substance abuse) and more psychosocial problems in adolescents [45].

Greater parental monitoring might decrease the engagement in non-constructive leisure activities [17]. There is little evidence on this connection; studies have linked the participation in supervised activities with the low risk of substance use [46,47], but the association between parental monitoring and non-constructive leisure should continue to be explored. No evidence has been found on the association between parental affection and a lower frequency of non-constructive leisure activities. These variables should also continue to be explored.

Our results also confirm that non–constructive leisure activities were lower among those with a high frequency of church attendance, but not among those with high frequency of praying or with high religious salience. A previous study had found that both attendance and salience predicted lower frequencies of unstructured leisure, although this association varied according to the country studied [35].

### 4.3. Substance Use

Our results are not clear with regard to the association between parental monitoring and substance use. Both in the regression and in the SEM, monitoring seems to be protective, but the effect is significant only in the SEM. These results would stand in the same protective direction observed in previous studies [26,27,47,48,49,50,51,52]. To try to explain this association, parental monitoring should be understood as a series of skills and behaviours of parents that make them aware of the activities of their children. This allows them to know their children’s friends and establish a relationship under an active process of communication with rules and norms based on parental knowledge [27,49,52,53]. Adolescence is a time in which young people want to spend more time with their peers and less time with their parents [27]. This change can increase the risk of substance consumption. Adolescents who have less parental monitoring may have greater opportunities to engage in substance use [53]; therefore, it can be said that parental monitoring can be a protective factor [27].

The results indicate that greater parental affection is significantly associated with a lower prevalence of substance use. Previous research has found similar associations between parental warmth and substance use [21,54,55,56]. This study provides new knowledge about the association between the two variables and covers various substances (alcohol, tobacco, marijuana and other drugs) in preadolescents. In this study, the parental affection measure offers more information about the family characteristics because it combines parent–child communication, relationship and support. These associations can arise because a positive and quality relationship with parents favours better psychological well-being in adolescents. A parent’s willingness to talk with adolescents about their concerns and problems offers the adolescents greater prominence and participation in family conversations [50,54,57] and is associated with a decrease in the adoption of risk behaviours.

Our results also confirm, in line with previous studies, that substance use is lower among women [58,59] and in young people who live with both biological parents [59,60,61].

With regard to the role of religiosity, church attendance predicted lower odds of substance use in the regression analyses, but prayer and salience did not. Previous studies have found results with similarities and differences with ours. Most of them found that church attendance is negatively associated with substance use [43,62,63,64,65]. Religious salience has been found to have less effect on substance use [62,65] or even no effect [43], which would be in accordance with our results. On the contrary, one study found that salience predicted lower substance use while attendance did not [35]. Prayer, unlike our results, has been found to predict lower substance use too [43,64]. However, when predicting other variables such as health and life satisfaction, spirituality, and not religiosity, was found to be a good predictor [66]. Future studies should test whether, when adjusting for spirituality, the role of religiosity on substance use decreases or disappears.

### 4.4. Family Assets: Parental Monitoring and Affection

Regarding our main variables, the results presented in this study, where we used the same questionnaire in four different countries, seem to confirm that the family can be a protective resource associated with positive youth development. Specifically, the results confirm the associations between parental monitoring and affection, and health assets such as the reduction and prevention of substance use and with greater time spent in constructive leisure activities. These findings are very much in line with previous results about the fundamental role that parents have in the decision-making of adolescents with respect to the establishment of certain lifestyles [26,49].

This study also found that three dimensions of the religiosity are associated with parental monitoring and affection. This would imply that religiosity would have a double effect on adolescent behaviour: a direct effect (as discussed above) and an indirect effect through the role of parenting. These associations should be further explored because recent studies found that only spirituality was associated with these parenting variables [67].

Few studies have included models that examined the associations between monitoring and parental affection [26,27,57] and activities linked to both the deficit model and the positive development model [27]. Most studies address only one type of outcome [48]. Recent studies have examined for example the associations of monitoring and/or parental affection or other family characteristics with substance use, unhealthy dietary habits, delinquent behaviour, sexual risk behaviours [26,27,48] and organised leisure activities [17] among adolescents. However, no other studies have previously examined the association between parental monitoring and affection and a broad range of lifestyle behaviours (constructive leisure, non-constructive leisure and substance use). This gap makes it difficult to apply existing knowledge in developing generic preventive approaches aimed at adoption of positive and healthy lifestyles taking into account the family as a protective resource.

Most studies have focused on European or North American populations. Some studies which include Hispanic preadolescents focus on risk behaviours such as drug abuse [68], delinquency and sexual risk [69], and they tend to use small samples [49]. Our study addresses this gap with a focus in an international sample of racially and ethnically diverse groups (Spain, Mexico, Chile and Peru) which includes a large number of Hispanic and European preadolescents (aged 12–15 years).

### 4.5. Limitations and Strengths

Different limitations should be taken into account when interpreting the results of this study. First, the results come from a cross-sectional study that does not necessarily allow for causal inferences to be made about the associations found. In cross-sectional studies inverse causality can explain some associations. For example, a low religiosity can determine riskier behaviours, but risky behaviours could also determine an adolescent to decrease their religiosity. However, inverse causality is not a likely explanation for the association between low monitoring and higher substance use. It is not likely that parents would lower their monitoring upon knowing that their adolescent is consuming drugs. The associations found could also be due to the role of confounding variables; however, we tried to adjust for confounders and had a good sample size to do so. In any case, longitudinal studies should confirm these results.

In addition, we used a convenience sample, and the response rate is unknown. However, this issue may have been partially compensated by our large sample size that enabled adjusted good adjustment for several potential confounders.

It should also be noted that, by filling in a self-completed questionnaire in which information was requested about their own perceptions, the students could have given answers that were socially well accepted and not so much their true perceptions. To avoid this possible bias, we insisted on anonymity, and we ensured that the responses were confidential, which reduces this kind of bias [70].

This study also has several strengths. First, the large sample size allowed the results to be generalised and allowed the analysis to be adjusted for various potentially confounding variables. In addition, the sample came from four underrepresented countries in the literature, which share some cultural traits but have significant differences between them. In addition, the fact that the main associations remained stable among the four countries gives strength to the generalisation of the results.

## 5. Conclusions

This study adds to the evidence that parental monitoring and affection may be one health asset leading to positive and healthy lifestyles. The results could help to take into account the family as a resource when designing health interventions in order to lead adolescents to participate in constructive leisure activities and avoid risky behaviours.

## Figures and Tables

**Figure 1 ijerph-17-08170-f001:**
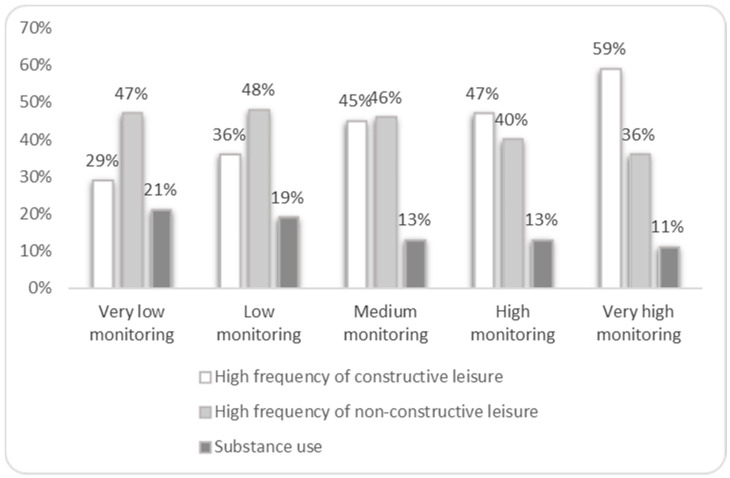
Constructive leisure, non-constructive leisure and substance use in adolescents, according to the degree of parental monitoring.

**Figure 2 ijerph-17-08170-f002:**
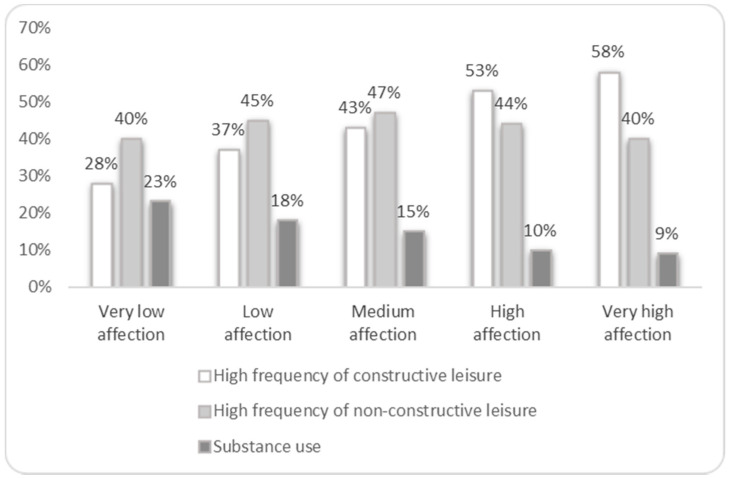
Constructive leisure, non-constructive leisure and substance use in adolescents, according to the degree of parental affection.

**Figure 3 ijerph-17-08170-f003:**
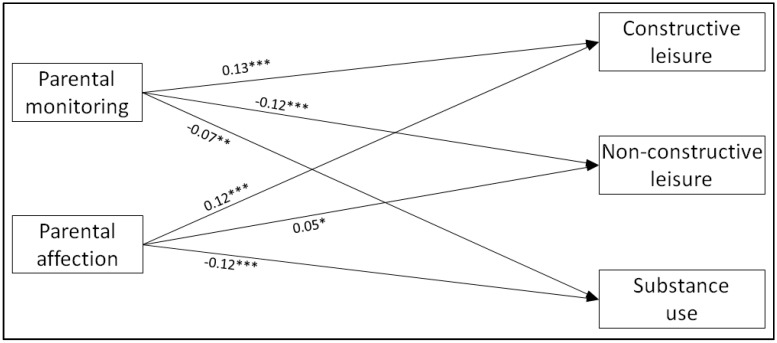
Structural equation model assessing the associations between parental variables and adolescent outcomes. Notes: Standardised estimates are shown. Age, sex and the three religiosity variables were included in the model (predicting both parenting and adolescent variables) but are not shown for the sake of clarity. Errors and items measuring the constructs are omitted for the same reason. *p*-values for the structural coefficients: * *p* < 0.05; ** *p* < 0.01; *** *p* < 0.001.

**Table 1 ijerph-17-08170-t001:** Characteristics of the sample.

Characteristics	Chile *n* (%) *N* = 393	Spain *n* (%) *N* = 964	Mexico *n* (%) *N* = 1089	Peru *n* (%) *N* = 854	Total *n* (%) *N* = 3300
**Sex**					
Male	172 (43.8)	413 (42.8)	568 (52.2)	337 (39.5)	1490 (45.2)
Female	221 (56.2)	551 (57.2)	521 (47.8)	517 (60.5)	1810 (54.9)
**Age (years)**					
12	11 (2.8)	9 (0.9)	103 (9.5)	109 (12.8)	232 (7.0)
13	183 (46.6)	630 (65.4)	483 (44.4)	510 (59.7)	1806 (54.7)
14	179 (45.6)	273 (28.3)	392 (36)	212 (24.8)	1056 (32.0)
15	20 (5.1)	52 (5.4)	111 (10.2)	23 (2.7)	206 (6.2)
**Two-parent household**					
No	111 (31.1)	88 (9.8)	151 (15.3)	163 (20.5)	513 (16.9)
Yes	246 (68.9)	806 (90.2)	838 (84.7)	632 (79.5)	2522 (83.1)
**Parents with a university education**					
No	81 (27.7)	121 (16.90)	218 (24.63)	184 (26.1)	604 (23.3)
Yes	211 (72.3)	595 (83.1)	667 (75.4)	521 (73.9)	1994 (76.8)
**Religion**					
No religion	132 (38.6)	181 (20.7)	85 (8.6)	66 (8.4)	464 (15.5)
Catholic	160 (46.8)	655 (75.0)	886 (89.2)	682 (86.3)	2383 (79.5)
Other	50 (14.6)	37 (4.2)	22 (2.2)	42 (5.32)	151 (5,0)
**Parental monitoring**					
Low	225 (65.4)	496 (56.9)	494 (50.4)	389 (49.3)	1604 (53.7)
High	119 (34.6)	376 (43.1)	486 (49.6)	400 (50.7)	1381 (46.3)
**Parental affection**					
Low	193 (56.8)	370 (44.2)	454 (47.6)	454 (61.8)	1471 (51.3)
High	147 (43.2)	468 (55.9)	500 (52.4)	281 (38.2)	1396 (48.7)
**Constructive leisure**					
Little	258 (69.4)	539 (57.8)	509 (48.7)	512 (61.7)	1818 (57.2)
A lot	114 (30.7)	394 (42.2)	536 (51.3)	317 (38.2)	1361 (42.8)
**Non-constructive leisure**					
Little	208 (55.9)	500 (53.7)	538 (51.5)	533 (64.6)	1770 (56.1)
A lot	164 (44.1)	432 (46.4)	506 (48.5)	292 (35.4)	1394 (43.9)
**Substance use**					
Never	286 (79.0)	737 (81.7)	845 (83.8)	730 (90.2)	2598 (84.0)
Some time	76 (21.0)	165 (18.3)	175 (17.2)	79 (9.8)	495 (16.0)

**Table 2 ijerph-17-08170-t002:** Variables associated with parental monitoring and affection.

		High Parental Monitoring				High Parental Affection		
	*N*	*n* (%)	*p* ^a^	OR (95% CI) ^b^	*N*	*n* (%)	*p* ^a^	OR (95% CI) ^b^
**Sex**								
Male	1338	670 (50.1)	<0.001	(ref)	1273	647 (50.8)	0.041	(ref)
Female	1647	711 (43.2)		0.82 (0.69–0.97)	1594	749 (47.0)		0.95 (0.80–1.13)
**Age**								
12–13	1855	884 (47.7)	0.051	(ref)	1777	865 (48.7)		(ref)
14–15	1130	497 (44.0)		0.81 (0.68–0.97)	1090	531 (48.7)	0.984	0.99 (0.83–1.18)
**Country**								
Spain	872	376 (43.1)	<0.001	(ref)	838	468 (55.85)	<0.001	(ref)
Chile	344	119 (34.6)		0.83 (0.61–1.14)	340	147 (43.2)		0.71 (0.52–0.96)
Mexico	980	486 (49.6)		1.13 (0.91–1.40)	954	500 (52.41)		0.80 (0.64–1.00)
Peru	789	400 (50.7)		1.25 (0.99–1.57)	735	281 (38.23)		0.47 (0.37–0.60)
**Two-parent household ^c^**								
No	476	186 (39.1)	<0.001	(ref)	449	161 (35.9)	<0.001	(ref)
Yes	2433	1169 (48.1)		1.11 (0.88–1.41)	2351	1209 (51.4)		1.38 (1.07–1.77)
**Parents with a University education**								
No	591	240 (40.6)	<0.001	(ref)	561	224 (39.9)	<0.001	(ref)
Yes	1938	949 (49.0)		1.12 (0.91–1.38)	1861	972 (52.2)		1.28 (1.03–1.58)
**Church attendance**								
None/Low	1548	590 (38.1)	<0.001	(ref)	1474	605 (41.0)	<0.001	(ref)
High	1397	772 (55.3)		1.44 (1.19–1.75)	1355	772 (57.0)		1.23 (1.01–1.50)
**Prayer**								
None/Low	1003	337 (33.6)	<0.001	(ref)	947	351 (37.1)	<0.001	(ref)
High	1934	1021 (52.8)		1.38 (1.11–1.70)	1873	1025 (54.7)		1.35 (1.08–1.67)
**Religious salience**								
None/Low	1084	381 (35.2)	<0.001	(ref)	1034	371 (35.9)	<0.001	(ref)
High	1798	961 (53.5)		1.62 (1.33–1.98)	1728	975 (56.4)		1.90 (1.55–2.32)

^a^*p* values of the bivariate χ^2^ tests. ^b^ Multiple logistic regression odds ratios (and 95% confidence intervals) of high supervision and affection, adjusted for all the variables in the table. ^c^ Two-parent household: refers to when the adolescent refers he/she lives with both parents. Ref, reference.

**Table 3 ijerph-17-08170-t003:** Variables associated with constructive leisure, non-constructive leisure and substance use.

	Constructive Leisure ^a^				Non-Constructive Leisure ^b^				Substance Use ^c^			
	*N*	*n* (%)	*p* ^d^	OR (95% CI) ^e^	*N*	*n* (%)	*p* ^d^	OR (95% CI) ^e^	*N*	*n* (%)	*p* ^d^	OR (95% CI) ^e^
**Monitoring**												
Low	1597	566 (35.4)	<0.001	(ref)	1597	755 (47.3)	<0.001	(ref)	1551	286 (18.4)	<0.001	(ref)
High	1377	712 (51.7)		1.49 (1.25–1.79)	1374	543 (39.5)		0.71 (0.60–0.86)	1357	167 (12.3)		0.84 (0.65–1.08)
**Affection**												
Low	1464	501 (34.2)	<0.001	(ref)	1464	643 (43.9)	0.605	(ref)	1432	283 (19.9)	<0.001	(ref)
High	1393	724 (52.0)		1.52 (1.26–1.82)	1392	598 (43.0)		0.94 (0.79–1.13)	1370	142 (10.4)		0.49 (0.38–0.64)
**Sex**												
Male	1432	625 (43.7)	0.390	(ref)	1424	668 (46.9)	0.002	(ref)	1393	267 (19.2)	<0.001	(ref)
Female	1747	736 (42.1)		1.09 (0.91–1.30)	1749	726 (41.5)		0.90 (0.76–1.13)	1700	228 (13.4)		0.73 (0.58–0.93)
**Age (years)**												
12–13	1965	827 (42.1)	0.293	(ref)	1960	789 (40.3)	<0.001	(ref)	1917	224 (11.7)	<0.001	(ref)
14–15	1214	534 (44.0)		0.98 (0.81–1.18)	1213	605 (49.9)		1.37 (1.14–1.63)	1176	271 (23.0)		2.24 (1.76–2.85)
**Country**												
Spain	933	394 (42.2)	<0.001	(ref)	932	432 (46.4)	<0.001	(ref)	902	165 (18.3)	<0.001	(ref)
Chile	372	114 (30.7)		0.72 (0.52–1.01)	372	164 (44.1)		0.79 (0.58–1.07)	362	76 (21.0)		0.78 (0.52–1.15)
Mexico	1045	536 (51.3)		1.48 (1.18–1.86)	1044	506 (48.5)		1.22 (0.97–1.52)	1020	175 (17.2)		0.83 (0.61–1.12)
Peru	829	317 (38.2)		0.82 (0.64–1.06)	825	292 (35.4)		0.64 (0.50–0.82)	809	79 (9.8)		0.49 (0.35–0.70)
**Two-parent household ^f^**												
No	510	175 (34.3)	<0.001	(ref)	511	221 (43.3)	0.780	(ref)	490	100 (20.4)	0.002	(ref)
Yes	2507	1119 (44.6)		0.94 (0.73–1.22)	2500	1098 (43.9)		0.97 (0.58–1.07)	2454	361 (14.7)		0.67 (0.49–0.92)
**Parents with a university education**												
No	602	180 (29.9)	<0.001	(ref)	601	189 (31.5)	<0.001	(ref)	587	88 (15.0)	0.328	(ref)
Yes	1985	966 (48.7)		1.92 (1.53–2.40)	1980	948 (47.9)		2.14 (1.72–2.67)	1947	325 (16.7)		1.37 (1.01–1.84)
**Church attendance**												
None/low	1594	527(33.1)	<0.001	(ref)	1594	704 (44.2)	0.735	(ref)	1544	285 (18.5)	<0.001	(ref)
High	1439	765(53.2)		1.39 (1.14–1.70)	1435	625 (43.6)		0.81 (0.66–0.99)	1415	189 (13.4)		0.72 (0.54–0.95)
**Prayer**												
None/low	1034	291 (28.1)	<0.001	(ref)	1031	451(43.7)	0.926	(ref)	994	207 (20.8)	<0.001	(ref)
High	1992	1003 (50.4)		1.54 (1.23–1.93)	1990	874 (43.9)		1.20 (0.96–1.50)	1957	267 (13.6)		0.84 (0.62–1.13)
**Religious salience**												
None/low	1119	360 (32.2)	<0.001	(ref)	1116	520 (46.6)	<0.05	(ref)	1076	224 (20.8)	<0.001	(ref)
High	1832	910 (49.7)		1.26 (1.02–1.56)	1831	774(42.3)		0.84 (0.68–1.03)	1806	237 (13.1)		0.80 (0.60–1.06)

^a^ Having a high frequency (over the median) of the following activities: “Playing sports, going to the mountains, etc.”; “Volunteering”; “Playing in or attending artistic and training activities”; and “Doing activities with your parents”. ^b^ Having a high frequency (over the median) of the following activities: “Hanging out in the street, in a park, at the beach, or in other public places”; “Going to shopping centres, game rooms, billiards, or football stadiums”; and “Getting together in a place with the group of friends, without adults present”. ^c^ Having ever used tobacco, alcohol, marijuana or other drugs. ^d^
*p* value for the bivariate χ^2^ test. ^e^ Multiple logistic regression odds ratios (and 95% confidence intervals) of each variable, adjusted for all variables in the first column. ^f^ Two-parent household: refers to the response that the adolescent gave about whether he/she lived with both parents. Ref, reference.

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
