# Peer review of "Connections between Family Assets and Positive Youth Development: The Association between Parental Monitoring and Affection with Leisure-Time Activities and Substance Use"

_ijerph, 2020, doi:10.3390/ijerph17218170_

Round 1

Reviewer 1 Report

The direct translation from Spanish to English encourages the use of excessively long sentences, which do not have the same meaning in English. Direct writing in English is recommended.

Introduction:The induction focuses on the characteristics of the population under study and on the origin of the research problem, but it would have to delve a little into the variables that make up said problem.

Method:The sample is sufficient and the general design adequate.

Instrument for collection data:The details of construction, validity and reliability of the instrument would be interesting to offer in the article. The variables that are considered in the instrument and the data that are collected on them are scarce to form such complex constructs as parental control or whether leisure is constructive or not.

Data analysis: A more direct cause-effect relationship between variables could be found by using a structural equation model.

Conclusions: The conclusions are excessively ambitious for the type of data collected and the analyzes performed.

Reviewer 2 Report

Manuscript ID: ijerph-979340
Type of manuscript: Article
Title: Connections between family assets and positive youth development: the
association between parental monitoring and affection with leisure-time
activities and substance use
Authors: Maider Belintxon, Alfonso Osorio *, Jokin de Irala, Marcia Van
Riper, Charo Reparaz, Marta Vidaurreta
Submitted to section: Health Behavior, Chronic Disease and Health Promotion,

I found the topic of the association between parental monitoring and affection with leisure-time activities and substance use in adolescents very interesting and worthwhile. The manuscript focuses more on positive adolescent development and not on a deficit-prevention model as most studies do. The manuscript is well written and focuses on the fact that family can be a protective resource associated with positive adolescent development characterised by a greater participation in constructive leisure activities and avoidance of risky lifestyles. My main comments are related to methodology.

I will continue with my remarks.

Methods

My first comment relates to the selection of the sample. Convenience samples choose the individuals that are easiest to reach. This type of non-probability sampling does not represent the entire target population so it is considered a bias. You should mention it in limitations.

You mention that invitation was sent out and 52 schools/3443 respondents participated but the information about the response rate is missing, that means how many schools/respondents were addressed. Please clarify.

Variables

My second comment relates to covariates. As you mention, religion was assessed as follows: They were initially asked what religion they belonged to. Then, if they belonged to any religion, they were asked how often they went to the church/temple of their religion and how often they prayed (from 0=Never, to 5=More than once a week). They were finally asked how much they agreed with this statement: “My faith is an important influence in my life, and I am willing to take it into account in my decisions”.

My suggestion is to assess separately questions about church attendance and praying and separately questions about the importance of faith and about the willingness to take into account faith when making decisions. The last two questions cover more internalised spiritual aspects of faith. Such separately assessing of church attendance and importance of faith was made in the paper of  Dankulincova Veselska, Z., Jirasek, I., Veselsky, P., Jiraskova, M., Plevova, I., Tavel, P., & Madarasova Geckova, A. (2018). International journal of environmental research and public health15(12), 2781.

My other question is if some measures of spirituality were used in YOURLIFE project. If there were some spirituality measures, I think you should reconsider spirituality as the covariate in the analyses. If not, my recommendation is to split the measurement of religion to church attendance / importance of faith.

Higher religiosity and spirituality is associated with higher parental monitoring but as for emotional support from parents, only spirituality was associated with higher emotional support. (Malinakova, K., Trnka, R., Bartuskova, L., Glogar, P., Kascakova, N., Kalman, M., ... & Tavel, P. (2019). Are adolescent religious attendance/spirituality associated with family characteristics?. International journal of environmental research and public health16(16), 2947.

Please, check the references, there are minor mistakes.

Round 2

Reviewer 1 Report

Authors are included all suggestions by reviewers. Therefore, I my opinion is ready to be considered for publication.

Author Response

Thanks a lot.

Reviewer 2 Report

Dear Authors,

I see discrepancies in the table 3, results and/or discussion. According to the table 3, church attendance, prayer and religious salience decreases the odds of substance use, but in results (page 9) and in discussion (page 13) you mention that only church attendance predicted lower substance use. Please revise.
